# Multi-Institutional Implementation of Clinical Decision Support for *APOL1, NAT2,* and *YEATS4* Genotyping in Antihypertensive Management

**DOI:** 10.3390/jpm11060480

**Published:** 2021-05-27

**Authors:** Thomas M. Schneider, Michael T. Eadon, Rhonda M. Cooper-DeHoff, Kerri L. Cavanaugh, Khoa A. Nguyen, Meghan J. Arwood, Emma M. Tillman, Victoria M. Pratt, Paul R. Dexter, Allison B. McCoy, Lori A. Orlando, Stuart A. Scott, Girish N. Nadkarni, Carol R. Horowitz, Joseph L. Kannry

**Affiliations:** 1Department of Medicine, Icahn School of Medicine at Mount Sinai, New York, NY 10029, USA; carol.horowitz@mountsinai.org (C.R.H.); joseph.kannry@mountsinai.org (J.L.K.); 2Department of Pathology, Icahn School of Medicine at Mount Sinai, New York, NY 10029, USA; 3Department of Medicine, Indiana University School of Medicine, Indianapolis, IN 46202, USA; meadon@iupui.edu (M.T.E.); emtillma@iu.edu (E.M.T.); prdexter@regenstrief.org (P.R.D.); 4Center for Pharmacogenetics and Precision Medicine and Department of Pharmacotherapy and Translational Research, College of Pharmacy, University of Florida Gainesville, Gainesville, FL 32610, USA; dehoff@cop.ufl.edu (R.M.C.-D.); nguyen.khoa@cop.ufl.edu (K.A.N.); marwood@trhc.com (M.J.A.); 5Division of Cardiovascular Medicine, College of Medicine, University of Florida, Gainesville, FL 32610, USA; 6Department of Medicine, Vanderbilt University Medical Center, Nashville, TN 37232, USA; kerri.cavanaugh@vumc.org; 7Department of Medical and Molecular Genetics, Indiana University School of Medicine, Indianapolis, IN 46202, USA; vpratt@iu.edu; 8Department of Biomedical Informatics, Vanderbilt University Medical Center, Nashville, TN 37232, USA; allison.b.mccoy@vumc.org; 9Center for Applied Genomics & Precision Medicine, Duke University School of Medicine, 101 Science Drive, Box 3382, Durham, NC 27708, USA; orlan002@duke.edu; 10Department of Genetics and Genomic Sciences, Icahn School of Medicine at Mount Sinai, New York, NY 10029, USA; sascott@stanford.edu; 11Department of Nephrology, Icahn School of Medicine at Mount Sinai, New York, NY 10029, USA; girish.nadkarni@mountsinai.org; 12Department of Population Health Sciences, Icahn School of Medicine at Mount Sinai, New York, NY 10029, USA

**Keywords:** *APOL1*, *NAT2*, *YEATS4*, clinical decision support, pharmacogenetics

## Abstract

(1) Background: Clinical decision support (CDS) is a vitally important adjunct to the implementation of pharmacogenomic-guided prescribing in clinical practice. A novel CDS was sought for the *APOL1*, *NAT2*, and *YEATS4* genes to guide optimal selection of antihypertensive medications among the African American population cared for at multiple participating institutions in a clinical trial. (2) Methods: The CDS committee, made up of clinical content and CDS experts, developed a framework and contributed to the creation of the CDS using the following guiding principles: 1. medical algorithm consensus; 2. actionability; 3. context-sensitive triggers; 4. workflow integration; 5. feasibility; 6. interpretability; 7. portability; and 8. discrete reporting of lab results. (3) Results: Utilizing the principle of discrete patient laboratory and vital information, a novel CDS for *APOL1*, *NAT2*, and *YEATS4* was created for use in a multi-institutional trial based on a medical algorithm consensus. The alerts are actionable and easily interpretable, clearly displaying the purpose and recommendations with pertinent laboratory results, vitals and links to ordersets with suggested antihypertensive dosages. Alerts were either triggered immediately once a provider starts to order relevant antihypertensive agents or strategically placed in workflow-appropriate general CDS sections in the electronic health record (EHR). Detailed implementation instructions were shared across institutions to achieve maximum portability. (4) Conclusions: Using sound principles, the created genetic algorithms were applied across multiple institutions. The framework outlined in this study should apply to other disease-gene and pharmacogenomic projects employing CDS.

## 1. Introduction

Clinical decision support (CDS) is an essential component underlying the successful integration of genomic data into clinical practice [1]. The National Human Genome Research Institute’s Implementing Genomics in Practice (IGNITE) consortium seeks to develop sustainable methods, including CDS, to support the utilization of genomic information in clinical care through a series of pragmatic clinical trials [2]. One such study, the pilot Genetic Testing to Understand and Address Renal Disease Disparities (GUARDD) study, investigated whether the return of *APOL1* genotyping results facilitated improvement in systolic blood pressure (SBP) at 3 months when compared with individuals who were not genotyped [3]. Given the positive preliminary results of the initial GUARDD trial, the study was expanded to a multi-center, randomized, controlled pragmatic trial called GUARRD-US. This multicenter trial will build upon its predecessor by examining whether genetic testing for the *APOL1* gene and pharmacogenetic predictors of antihypertensive medication response will influence blood pressure control and antihypertensive medication selection in an African American (AA) hypertensive population. However, a key challenge to surmount for the success of the trial is a lack of an available CDS to support either *APOL1*-genotype or pharmacogenomic-guided therapy.

Significant evidence links the presence of *APOL1* high risk alleles to the excess prevalence of chronic kidney disease (CKD) in individuals of African ancestry [3]. Homozygosity for these risk alleles confers an approximately 2-fold higher risk for progression to end stage renal disease (ESRD) [4,5,6]. Analogous to outcomes in the general population of the SPRINT trial, strict blood pressure control has been associated with reduced mortality in AAs with *APOL1* risk alleles [7,8]. While renal outcomes such as CKD progression have not been consistently improved by strict BP control [6,9], there is preliminary evidence that individuals with *APOL1* risk alleles will have greater blood pressure response and albuminuria reduction with renin-angiotensin-aldosterone system (RAAS) antagonism than other antihypertensive agents [10]. The clinical indications for *APOL1* testing and the actionability of *APOL1* results are the subject of ongoing investigation [11]. The GUARDD-US study seeks to fill a knowledge gap, as consensus guidelines for clinical decision support are not yet widely available.

Additionally, the multi-center GUARDD-US study explores whether genetic differences result in differential responses to antihypertensive therapy in AAs. Significant preliminary data support the association of two drug–gene pairs employed in this pragmatic trial. The first is an expression quantitative trait locus linking YEATS4 to thiazide diuretic efficacy [12,13,14]. The second is a pharmacokinetic drug–gene pair, wherein N-acetyl-transferase 2 (*NAT2*) genotype confers differential drug levels and response to hydralazine [15]. Although guidelines from the clinical pharmacogenomics implementation consortium have not been curated for antihypertensive agents, pharmacogenomics (PGx) testing has the potential to inform both initial antihypertensive agent selection and those who might be resistant to specific antihypertensive medications. To enable multi-site implementation of *APOL1*, *YEATS4* and *NAT2* CDS in an electronic health record (EHR), the CDS committee assembled to translate clinical content into discrete algorithms. The approach undertaken in this project necessitated unification and standardization of CDS across multiple sites, with a focus on implementation in the EpicCare Ambulatory EHR platform.

## 2. Materials and Methods

Five clinical groups consisting of 10 recruiting sites implemented CDS for genotype-guided therapy in clinical use (Table A1). This was in preparation for the multi-center, randomized, controlled pragmatic trial GUARRD-US (IRB: Pro00102979). The majority of sites subscribed to the EpicCare Ambulatory EHR. One collaborating site utilized Cerner solutions EHR. Therefore, the development of the CDS system was focused on EpicCare deployment; however, the principles and logic were developed to allow portability to other EHR platforms. 

A unified CDS committee consisting of clinical content experts and clinical decision support experts was formed to develop an integrated approach with resulting cohesive clinical and EHR-based algorithms. This helped us to achieve a better understanding of study content and decisions, coupled with an understanding of what CDS could support and how CDS could meet content requirements. It was ensured that the proposed clinical study algorithms for *APOL1* and antihypertensive pharmacogenetics could easily be translated in application for EHR use. The committee included research principle investigators (as needed), clinical providers, pharmacists, and IT analysts. As needed, the authors J.K. (committee chair) and T.S. interfaced with both the lab directors and the trial coordinating center responsible for database development and randomization. 

The GUARRD CDS committee were guided by both general CDS and project-specific principles (see Table 1). For the CDS principles, an overall sound clinical study algorithm rooted in the latest scientific knowledge was sought. Most of the CDS principles were derived from an earlier framework to successful CDS [16]. 

CDS implementation framework was built under the guidance of 4 general principles (Table 1): (1) medical algorithm consensus; (2) actionable; (3) context-sensitive triggers; and (4) workflow integration. First, algorithms must be medically reasonable and valid with broad provider acceptance, and as such the algorithms implemented in this study were vetted by the content experts whose justification is beyond the scope of this work. Second, CDS, where possible, should be actionable. An actionable alert is defined as one in which a direct diagnostic or therapeutic intervention could be derived from the alert recommendation. For example, recommendations should be crafted to tell providers how to adjust antihypertensive therapy selection, not simply advise them of inefficacy without providing an alternative therapy. 

Third, context sensitivity is ensured by using medical information in the electronic medical record to guide alert actions, such as designing smart sets that reference the patient’s medication and blood pressure status. Fourth, alerts were required to fit into the existing clinical workflow. Context-sensitive triggers are defined as workflow events that result in CDS intervention [16] For example, when cautioning providers not to prescribe a medication, an alert was placed within the prescription before the prescription could be placed. The 3rd and 4th principle are interrelated and differ slightly in that the 3rd principle is more focused on the triggers to support the devised medical algorithm and the 4th principle is largely focused on the specific timing of interventions.

Four project-specific derived principles which complemented the general principles included: (5) feasibility (6) interpretability (7) portability and (8) discrete result reporting. If a proposal was not technically feasible in the native EHR, it would not be considered further. Feasibility is defined as clinical study content recommendations which can be made functional in EHR-generated CDS and reflect the intent of clinical content. External CDS web services would be difficult if not impossible in IGNITE because of different levels of support at each clinical site. 

To achieve interpretability (principle 6), all interventions needed to be designed with specific instructions that could be readily understood by all levels of providers. Portability (principle 7) is primarily defined as being adaptable across multiple sites with a similar EHR. Sites with the same EHRs may have differences that do not always allow a perfect one to one conversion. Portability across EHRs was not the major focus, given that only a single site did not utilize EpicCare. 

Discrete result reporting (principle 8) was essential in making all of the prior principles possible. By discrete results, we mean only short succinct and finite observation values consisting of “G1/G1”, “G1/G2”, “G2/G2” for an APOL1 genotype being discretely sent through an HL7 interface in the OBX segment in addition to usual full text reports. It was discovered early on that genetics laboratories at various institutions did not always consistently provide discrete result components for their various tests, as most prefer large text reports. However, for automated CDS interventions to work using the native functionality of the EHR, discrete result components were necessary and heavily encouraged. 

As mentioned above, discrete results components were a key project principle identified to achieve the principle of feasibility. As such, regular meetings with the laboratory leadership of the representative sites were created to ensure that the laboratory understood the necessary requirements for discrete lab components and produced them in an automated consistent fashion to ensure compatibility with the designed CDSS. For example, placing values such as “G1/G1” in free text fields prone to user input error and deviations from the expected genotype values would not be interpretable by CDSS. 

## 3. Results

At the lead institution, six different CDS alert types were created for *APOL1*, four alerts for *YEATS4*, and eight alerts for *NAT2*. These alerts strived toward maintaining each general and project-specific principle outlined. An in-depth discussion is included for both the *APOL1* and *YEATS4* algorithms. Nearly identical execution of principles was performed on the *NAT2* algorithm; thus, a detailed discussion of the *NAT2* algorithm is given in the Appendix A. 

### 3.1. APOL1 Genotype-Guided Interventions

The goal of the *APOL1* CDS was to inform providers of the need to screen their patients for CKD and initiate/titrate RAAS inhibition when appropriate. Embracing the principle of actionability, a CDS alert is only applicable if the patient’s *APOL1* genotype is high risk (i.e., *APOL1* positive). Additional criteria that differentiated the type of CDS alert received by a provider included the presence of uncontrolled hypertension and an increased level of or lack of a recent albumin/creatine ratio. The overall algorithm for *APOL1* genotype-guided therapy is described in Figure 1 and each alert is described below:One Time Notification Alert∘General for your information (FYI) alert (no specific criteria) that triggers once, informing the clinic of the patient’s high-risk phenotype for CKD and the need for annual screening.*APOL1* No Albumin/Creatinine Ratio Alert
∘This alert triggers at every encounter if there is no albumin/creatinine ratio result. It tells the provider that ordering an albumin/creatinine ratio is recommended due the patient’s high-risk allele.*APOL1* HTN Normal Ratio Alert
∘Triggers at every encounter when the albumin/creatinine ratio is normal but the patient’s blood pressure is high. It then provides general blood pressure control recommendations.*APOL1* NOT On RAAS Antagonist HTN Alert
∘This alert triggers at every encounter when the patient is (1) not on a RAAS antagonist (2) has a blood pressure that is >140/90 and (3) has an increased albumin/creatinine ratio. It then recommends that the provider add RAAS antagonist therapy to control HTN.*APOL1* NOT On RAAS Antagonist Proteinuria Alert
∘This alert triggers at every encounter when the patient is (1) not on a RAAS antagonist and (2) has a blood pressure that is <140/90 but greater than >130/80. It then recommends that provider to add RAAS antagonist therapy to prevent proteinuria.*APOL1* On RAAS Antagonist Modification Alert
∘This alert triggers at every encounter if the patient is (1) on a RAAS antagonist, (2) has a blood pressure that is >130/80, and (3) has an increased albumin/creatinine ratio. It then recommends the provider to modify the current RAAS antagonist therapy.

Adhering to the first general principle, the CDS committee achieved a general consensus regarding clinical recommendations. A clear guideline set forth by the members of the trial was essential for the success of the CDS intervention. 

For the *APOL1* genotype-guided therapy, workflow principles 3 and 4 (see Table 1) were difficult to implement. A trigger could not be devised that occurred when a provider is looking at a laboratory result or is thinking about modifying the general blood pressure medications of the current patient. Fortunately, in the Epic EHR, there exists a general CDS capability in which providers can receive a general CDS alert when a specific trigger alert cannot be created. For many institutions in this trial, this section has become part of the usual ambulatory provider workflow. Providers are encouraged to review these alerts at every encounter. Therefore, while a specific trigger could not be created, there was at least some level of workflow integration for most institutions. At sites without Epic, however, general alerts upon opening of the chart needed to be implemented.

Figure 2 demonstrates an example CDS alert. This alert was designed to be both actionable and interpretable. All of the information necessary for the provider to digest the information and take the appropriate action is available on the alert. The patient’s most recent blood pressure and microalbumin ratio is displayed. If the provider and/or patient is interested in further information, both provider- and patient-friendly handouts are available as hyperlinks. The alert also contains a link to an orderset with orders for common dosages of lisinopril (5, 10, 20, and 40 mg) and losartan at three common dosages (25, 50, and 100 mg). Institutions were allowed the flexibility to modify the orderset to meet each institution’s policies and formulary.

The problem and reason for the alert is at the top and begins with an emboldened and underlined “Problem” heading. After a short succinct sentence, the alert moves on to an emboldened and underlined recommendation. All these design choices were made to help providers obtain the information they need quickly without needing to access another tab or windows and allow providers to take action on the spot. Recommendations had clear instructions as to the suggested actions to take and were designed to be direct but at the same time not overtly forceful. Even the FYI alerts were actionable in that they instructed providers to take action by informing their patients of an increased risk of CKD. 

In the first iteration of the CDS alert, the recommendation stated treatment with a “*RAAS* antagonist” is necessary. After some discussions with stakeholders, the CDS committee concluded that this language was unnecessarily complex as general providers may not immediately associate *RAAS* antagonists with an ACE-inhibitor or angiotensin receptor blocker. This experience highlights the need for diverse stakeholder input on the language set forth by the CDS alerts to achieve maximum interpretability.

In the CDS design, there were acknowledged reasons, but some of these acknowledged reasons did not affect the lockout period for the best practice alerts (BPAs). An example of this would be the albumin/creatine ratio, as it would reappear in every encounter no matter the response. The CDS committee thought that this was fair due to the medical necessity and the fact that the CDS alerts were already restricted to only be shown to certain provider specialties in the ambulatory setting.

### 3.2. Hydrochlorothiazide and YEATS4

The overall CDS algorithm for *YEATS4* is highlighted in Figure 3. First line antihypertensive therapy in AAs typically consists of either a calcium channel blocker or thiazide diuretic. The presence of a TC or TT genotype predicts reduced *YEATS4* expression and reduced thiazide efficacy. In this circumstance, the algorithm alerts providers to the risk of thiazide inefficacy and preferentially suggests calcium channel blocker selection as the initial antihypertensive therapy. For *YEATS4*, there is only an intervention if a patient has a genotype associated with decreased expression and efficacy. If the patient has the decreased expression phenotype and is not on a calcium channel blocker, then a CDS alert appears if (1) a thiazide diuretic is on the medication list or (2) the provider starts to order a thiazide diuretic for treatment of hypertension (HTN). If a thiazide diuretic is on the medication list, then the CDS report will appear in the general CDS section. When a provider places an order but has yet to complete the medication order form, a pop-up alert will appear. An alert should fire once per encounter when a physician tries to order a thiazide medication for HTN. In addition, the general alert in the CDS section will fire every encounter if the patient has a blood pressure >140/90.

The HTN thresholds for *APOL1* were ported over to *YEATS4* and a general medical consensus of the algorithm (principle 1) was easily achieved. The language for the alert was similar to the *APOL1* alert, thus achieving a high degree of actionability (principle 2) and interpretability (principle 6). First iterations of the CDS alert used language such as “this patient’s *YEATS4* genetic test is associated with reduced thiazide efficacy.” However, in order to achieve maximum interpretability, the language was changed to “reduced hydrochlorothiazide efficacy and possible reduced chlorthalidone efficacy.” Electing to specifically spell out the affected medication was better received by the various stakeholders.

To ensure actionability, the CDS alert contained an orderset for amlodipine as the suggested calcium channel blocker with dosage options of 5 and 10 mg. It was emphasized for portability that the recommended calcium channel blocker and dosage be modified to fit each institutions formulary. 

The medication interruptive alert was fine-tuned to help achieve ideal workflow integration (principle 4) by launching after the provider selects a hydrochlorothiazide or chlorthalidone order but before the medication order form appears. In addition, a condensed message in the medication order form itself was placed, achieving another degree of workflow integration. 

### 3.3. Hydralazine and NAT2

Further discussion on the *NAT2* algorithm is reserved for the Appendix A. However, a discussion on the whether the medical algorithm should contain a decision point for the presence or absence of resistant hypertension is warranted. Having a decision point for the presence or absence of resistant hypertension means the naïve EHR functionality would need to be able to detect this patient population. This was found by the committee to not meet the feasibility principle. The current definition of resistant hypertension by the 2018 AHA scientific statement is blood pressure that remains above the goal in spite of concurrent use of three antihypertensive agents of different classes. An up-to-date list of medications classified specifically as hypertension medications actively maintained by reputable source was not available to the CDS committee’s knowledge. Therefore, it would require the CDS committee to go through all of the possible hypertension medication themselves and this would require a significant amount of maintenance and upkeep. The solution was therefore to modify the verbiage to make it clear that the recommendations in the alert were for patients with resistant hypertension only.

### 3.4. Portability to Other Sites

The EHR build was developed to achieve the principle of maximum portability across sites with the same EpicCare EHR. At this time, an importation application with tools and features that would allow an analyst to reconcile differences in build requirements for various EHRs was not available for EpicCare. Therefore, the committee had to report to rudimentary knowledge sharing of build specifications by providing all sites a PowerPoint with screenshots of all the different parameters needed to be entered by analysts in the respective build environment in the EHR. Using provided documents, the collaborating sites have succeeded in replicating the CDS algorithms built at the lead site. 

### 3.5. Other Sites Experience

Across the consortium, each institution was queried for their CDS implementation experience, as slight modifications in the algorithm designed by the CDS committee and beta tested at Mount Sinai may have been necessary. At Indiana University, their safety-net hospital subscribes to the EpicCare EHR and has a rich history of CDS innovation [17]. The IT support team was able to implement the developed CDS algorithms seamlessly without alterations of the primary site’s protocol. It should be noted that Indiana’s University hospital runs the Cerner Solutions EHR system and has achieved functional implementation equivalence of CDS BPAs with minimal alterations. 

At the University of Florida (UF), Epic’s EMR was successfully rolled out in 2010. The UF Personalized Medicine Program has experience implementing many pharmacogenetics CDSs for both clinical care and research [18]. The UF Pathology Lab also has experience providing genetic values in discrete results to help with Epic’s alert system. Optional one-time notifications and alerts providing a condensed message in the medication order form were not included. Due to a lack of targeted identification of patients with resistant HTN for *NAT2* PM, alerts in the general CDS section were excluded. All BPAs were approved by local CDS governance committee and implemented at two hospitals: Shands and Jacksonville. 

At Vanderbilt University Medical Center, the decision was made to trigger the CDS using genomic indicator within EpicCare for *APOL1* to eliminate the need for a new laboratory interface. Genomic indicators function as tags that are created by an administrator. These tags can then be added to a patient’s chart by a clinician to indicate possible genetic risks for certain drugs or diseases. When the study team receives the laboratory result, it is uploaded into REDCap and imported as a PDF into Epic. For patients who are *APOL1* high risk positive, a study clinician then adds a *APOL1* high risk for CKD genomic indicator to the patient’s chart, making it visible to clinicians and patients and possible to trigger CDS. The local team opted to include the one-time notification alert but combined it with the alert for *APOL1*-positive patients with a normal albumin/creatinine ratio and hypertension, which provided the same general recommendation and did not present follow-up actions. Otherwise, the CDS build was straightforward using the shared screenshots, and only minor changes were made to localize site-specific identifiers (e.g., departments, laboratory test results, and medications) and to conform to local style guidelines (e.g., avoiding negative logic in alert criteria and creating a grouper to restrict the site to study departments).

Meharry Medical College, Southeastern Healthcare, ‘Baylor, Scott, and White (Baylor)’, University of North Carolina at Pembroke (UNC-P), and University Medical Center- New Orleans (UMC-NO) chose to manually deploy alert content due to limited resources and support not conducive for implementation. The associated research coordinator reviews *APOL1* test results and then manual sends alert content to providers for patients who are *APOL1* high risk positive and meet other context sensitive criteria for an alert. These health systems chose to deploy the *APOL1* alerts only and not the *NAT2* or *YEATS4* alerts. 

## 4. Discussion

PGx CDS implemented at a single institution utilizing multiple genes have been previously described [19]. However, this study looks at a large multi-institutional attempt to develop PGx CDS for a small number of genes [20]. General frameworks for CDS have previously been applied to genomic medicine with success [1]. Similarly, the proposed framework is an amalgamation of earlier frameworks from a general CDS standpoint [1,16,21]. Utilizing the framework, the uniform CDS committee created novel algorithms in PGx for a small number of genes.

A novel algorithm for *APOL1*, *NAT2*, and *YEATS4* genetic testing has been created, translating the clinical algorithms into something programmable with maximum portability while being mindful of key guiding principles throughout the process. This is the first medical and CDS algorithm for *APOL1*, a key risk factor in the incidence of CKD for AAs. Utilizing the patients’ most recent albumin/creatinine ratio and blood pressure, purposeful alerts guide the provider into taking appropriate action. These alerts contain ordersets for typical RAAS inhibition with suggested doses that vary depending on the circumstance, clear links to provider and patient handouts, and pertinent laboratory and blood pressure values. In most cases, the alert was placed in a general BPA section, a commonly utilized section in the ambulatory setting, to ensure efficient workflow integration.

This is one of the first medical and CDS algorithms developed for *YEATS4* and *NAT2*, two genes with supportive evidence for an association with hypertensive medications [22]. Both algorithms allowed for significant workflow integration by the alerting physicians during the initial ordering of a pertinent medication, well before the signing of an order. All of the alerts were succinct and concise with a clear statement of the purpose and the recommended actions to take. Multiple rounds of discussion of the language in the alerts with committee members and other stakeholders helped to maximize readability.

A clear, actionable CDS is essential for large multicenter trials introducing novel medical algorithms in the larger community. Prior studies have demonstrated a relative unease by primary care providers in incorporating genomic information in their practice [23]. Sauver et al. reported that 53% of primary care providers had a negative perceptive of pharmacogenetic alerts [24], with Olander et al. reporting that 76% of primary care providers are uncomfortable applying results of a pharmacogenetic test [25]. Some sites have experimented with having in-house vs. on-call pharmacists to deal with the potential questions that may arise because of pharmacogenetic testing [26]. Given the current comfortable level with genetic testing, CDS algorithms for pharmacogenetic testing must adhere to common CDS principles to maximize acceptance.

The major drawback of the *NAT2* algorithm was not being able to detect resistant hypertension automatically. The development and maintenance of a list of antihypertensives was the major roadblock. However, if this pragmatic trial shows successful results, sites may have significant rationale in eliciting further IT support in the development and maintenance of such a list.

The GUARDD trial is currently underway at the time of this manuscript. At the trial’s end, it is hoped to elicit some degree of provider feedback on the CDS interventions as well the overall adoption and acceptance rate. Monitoring the use of any implemented CDS is an important aspect in all CDS life cycle [27]. However, significant participation from multiple stakeholders was sought and thus the CDS committee is confident that the established framework will help achieve maximum rates.

For other sites thinking of implementing PGx CDS, the formation of a unified CDS committee consisting of clinical content (e.g., domain) experts and clinical decision support experts is highly recommended and was instrumental in the success of the developed CDS algorithm. In addition, streamlined governance for decision making helped progress the project efficiently. We have uploaded provider and patient handouts utilized in this clinical trial in the IGNITE toolbox located at: https://dcricollab.dcri.duke.edu/sites/NIHKR/Pages/IGNITEToolbox.aspx, accessed 26 May 2021.

## 5. Conclusions

In conclusion, we sought to create a simple, actionable novel CDS for *APOL1*, *NAT2*, and *YEATS4*. Through sound principles, a CDS with a high probability for success was achieved. This paper can serve as a roadmap for implementation at other institutions seeking to implement PGx-CDS at their respective institutions.

## Figures and Tables

**Figure 1 jpm-11-00480-f001:**
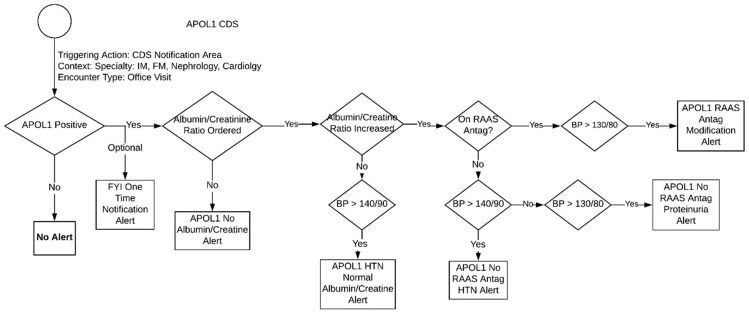
Proposed *APOL1* genotype guided therapeutic algorithm.

**Figure 2 jpm-11-00480-f002:**
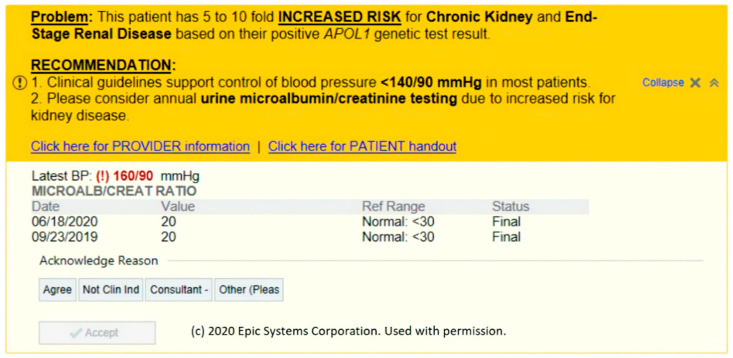
An example alert for *APOL1* in response to an elevated BP with a normal albumin/creatinine ratio.

**Figure 3 jpm-11-00480-f003:**
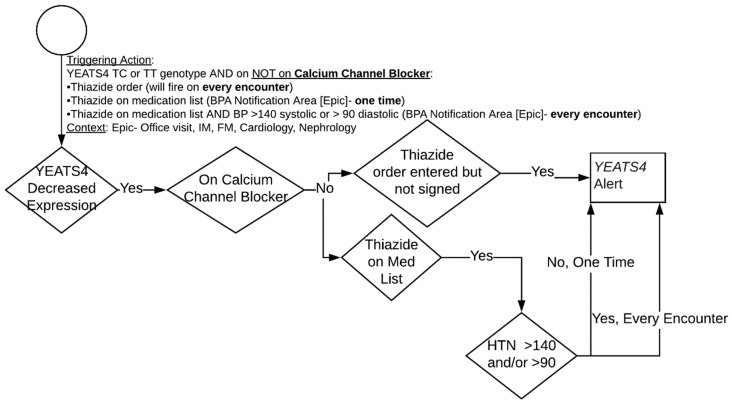
*YEATS4* genotype guided therapeutic algorithm.

**Table 1 jpm-11-00480-t001:** Guiding principles for the CDS committee.

CDS General Principles	CDS Project Specific Principles
1. Medical algorithm consensus	5. Feasibility
2. Actionability	6. Interpretability
3. Context-sensitive triggers	7. Portability
4. Workflow integration	8. Discrete Results Reporting

## Data Availability

The IGNITE toolbox is a publically available resource for resources applicable to genomic implementation. The toolbox is located at https://dcricollab.dcri.duke.edu/sites/NIHKR/Pages/IGNITEToolbox.aspx, accessed 26 May 2021. The resource documents will be uploaded to the toolbox after Network agreement on the addition of appropriate messaging and instructions for implementers who wish to utilize these tools.

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
