# Peer review of "Multi-Institutional Implementation of Clinical Decision Support for APOL1, NAT2, and YEATS4 Genotyping in Antihypertensive Management"

_jpm, 2021, doi:10.3390/jpm11060480_

Round 1

Reviewer 1 Report

This paper describes a novel CDS support for the APOL1, NAT2, and YEATS4 genes to guide pharmacotherapy of antihypertensive drugs among African Americans at multiple institutions. CDS is essential in implementing pharmacogenomic data into clinical practice and here the authors present one specific example of how we can develop a highly reliable CDS by creating genetic algorithms and applying these into the trial.

This work seems significant and scientifically sound. I only would like to comment on few details that must be corrected/improved:

Line 43: Build documents?

Line 56: when compared to → when compared with

Line 94: Cerner solutions EHR. What is Cerner? Do you mean Center?

Line 111 and 127: (2) Actionable → Actionability (parallelism with 3 other principles!)

Line 127: CDS Project Specific Principles → CDS Project-Specific Principles

Line 135: needed to be easily to be designed → needed to be easily designed

Line 138: differences that don’t always allow a perfect one to one conversion →

differences that do not always allow a perfect one-to-one conversion

Line 139: only a single site did not utilized EpicCare. → only a single site did not utilize EpicCare.

Line 156: APOL1 genotype guided interventions → APOL1 genotype-guided interventions

And put only one space between two flanking sentences. Not two (found all over the manuscript)! That is grammatically incorrect.

Author Response

Thomas Schneider, MD

425 So. Euclid Ave., Campus Box 8118

St. Louis, MO 63110

schneider.t@wustl.edu

schneiderthomas@gmail.com

517-896-1774

May 20, 2021

Dear Lola Liu and the Editorial Staff of JPM,

On behalf of the co-authors, we would like to re-submit our original manuscript entitled “Multi-institutional implementation of clinical decision support for APOL1, NAT2, and YEATS4 genotyping in antihypertensive management” for publication as an original research article in JPM.

We would like to thank the reviewers for their valuable comments. We have addressed all the reviewers’ suggestions and comments. We believe the revised manuscript is significantly enhanced by the feedback and remains of great interest to the readers of JPM. Please see below for a discussion of the improvements facilitated by your review organized by reviewer comments.  We also have attached the manuscript with tract changes to show the adjustments made due to comments.

Sincerely,

Thomas Schneider

Joseph Kannry

Reviewer 1:
This work seems significant and scientifically sound. I only would like to comment on few details that must be corrected/improved:

Thank you for your thorough read of the manuscript. We addressed your critiques below.

Line 43: Build documents? We apologize if this cause confusion.  This is in reference to the Powerpoint with screenshots discussed in section 3.4 (line 293).  These Powerpoints contained screenshots and instructions for other sites to implement the clinical decision support in their Epic environment to achieve maximum portability of our designed clinical decision support.  This was shortened due to word limitations in the abstract.  We have changed the term from “build documents” to “Implementation instructions.”

Line 56: when compared to → when compared with. Thank you for noticing this, we have corrected this grammatical mistake in the document.

Line 94: Cerner solutions EHR. What is Cerner? Do you mean Center? Cerner is an Electronic Health Record (EHR) Vendor. The full name of the company is Cerner Solutions

Line 111 and 127: (2) Actionable → Actionability (parallelism with 3 other principles!) Thank you, we have corrected this. 

Line 127: CDS Project Specific Principles → CDS Project-Specific Principles Thank you, we have corrected this.

Line 135: needed to be easily to be designed → needed to be easily designed Thank you, we have corrected this.

Line 138: differences that don’t always allow a perfect one to one conversion →

differences that do not always allow a perfect one-to-one conversion Thank you, we have corrected this.

Line 139: only a single site did not utilized EpicCare. → only a single site did not utilize EpicCare. Thank you, we have corrected this.

Line 156: APOL1 genotype guided interventions → APOL1 genotype-guided interventions Thank you, we have corrected this.

And put only one space between two flanking sentences. Not two (found all over the manuscript)! That is grammatically incorrect. Thank you. We have corrected this.
Reviewer 2 Report

Thank you very much and good luck for your paper

Author Response

Thomas Schneider, MD

425 So. Euclid Ave., Campus Box 8118

St. Louis, MO 63110

schneider.t@wustl.edu

schneiderthomas@gmail.com

517-896-1774

May 17, 2021

Dear Editorial Staff of JPM,

On behalf of the co-authors, we would like to re-submit our original manuscript entitled “Multi-institutional implementation of clinical decision support for APOL1, NAT2, and YEATS4 genotyping in antihypertensive management” for publication as an original research article in JPM.

We would like to thank the reviewers for their valuable comments. We have addressed all the reviewers’ suggestions and comments. We believe the revised manuscript is significantly enhanced by the feedback and remains of great interest to the readers of JPM. Please see below for a discussion of the improvements facilitated by your review organized by reviewer comments.

Sincerely,

Thomas Schneider

Joseph Kannry

Reviewer 2:

Comments and Suggestions for Authors
Thank you very much and good luck for your paper
Thank you for your review and input.

Reviewer 3 Report

The authors describe the implementation of PGx CDS for APOL1, NAT2, and YEATS4 genotyping in antihypertensive management in a multi-center setting spanning ten recruiting sites. The manuscript includes the decision tree of the algorithms and descriptions of the different alert types that were implemented, which could be of interest to other implementers. 

The manuscript is generally clearly written and understandable, however, the following improvements could be made to add value: 

  • (1) Include additional practical information for implementers.
  • (2) Include information on why several sites chose not to instead deploy manual solutions even though the design of the CDS was adapted to their EHR.
  • (4) Add a paragraph that frames the implementation in relation to other PGx CDS implementation projects and frameworks.
  • (3) Improve figures. Currently the font size is too small and the resolution is low.

More specific comments:

  • The majority of participating sites were using similar EHRs (EpicCare Ambulatory EHR) whereas one site was using Cerner solutions. The authors write that the designed CDS was therefore adapted to EpicCare, while the site usings Cerner Solutions did not yet achieve functional implementation of CDS BPA. Further, several sites using EpicCare Ambulatory chose to manually deploy alert content rather than using Epic. What were the reasons for this?
  • The manuscript would provide additional value for implementers if more concrete guidance could be given. Some examples include:
    • The authors emphasise the importance of discrete lab results reporting (p. 4). What does this specifically refer to and how was this implemented by the labs? Can you give guidance on this to other implementers who might face the same issue?
    • Are there any open source tools or resources available that could be of use to other implementers? E.g., the patient handouts?
  • The discussion would benefit from framing the implementation in relation to other PGx CDS implementation projects and frameworks.
  • The font size of Figure 1 and Figure 3 is a bit too small. At the top of Figure 3, ‘Text’ should be removed?
  • Generally, the resolution and appearance of the figures could be improved. For Figure 3, some text is overlapping (‘Thiazide order entered…’).
  • The abbreviation ‘HTN’ is used in the figures and text but not explained.

Author Response

Thomas Schneider, MD

425 So. Euclid Ave., Campus Box 8118

St. Louis, MO 63110

schneider.t@wustl.edu

schneiderthomas@gmail.com

517-896-1774

May 17, 2021

Dear Lola Liu and the Editorial Staff of JPM,

On behalf of the co-authors, we would like to re-submit our original manuscript entitled “Multi-institutional implementation of clinical decision support for APOL1, NAT2, and YEATS4 genotyping in antihypertensive management” for publication as an original research article in JPM.

We would like to thank the reviewers for their valuable comments. We have addressed all the reviewers’ suggestions and comments. We believe the revised manuscript is significantly enhanced by the feedback and remains of great interest to the readers of JPM. Please see below for a discussion of the improvements facilitated by your review organized by reviewer comments.

Sincerely,

Thomas Schneider

Joseph Kannry

Reviewer 3:

(1) Include additional practical information for implementers.

This is a great idea. We agree and have added a description that highlights the practical aspects for implementation. Specifically, we added to the discussion that it is necessary for having content and informatics experts in the same room and streamlining governance for decision making (see lines 385-390).  In addition, we have created a separate document that highlights the overall design for each created alert in which implementers may use as a template to help create their own clinical decision support alerts.  We will be uploading this document as well as provider and patient handouts utilized in the GUARDD trial after Network agreement on the addition of appropriate messaging and instructions for implementers who wish to utilize these tools. The general location for the IGNITE toolbox is below: https://dcricollab.dcri.duke.edu/sites/NIHKR/Pages/IGNITEToolbox.aspx

(2) Include information on why several sites chose not to instead deploy manual solutions even though the design of the CDS was adapted to their EHR.

Thank you for this comment. The rationale for not implementing CDS at all sites was a resource issue. Some sites did not have an EHR and resources conducive for implementation. We have added this to the paper (lines 337 - 338).

(3) Improve figures. Currently the font size is too small and the resolution is low.

Thank you for this comment. We have improved the resolution and increased the font size of the figures.

(4) Add a paragraph that frames the implementation in relation to other PGx CDS implementation projects and frameworks.

Thank you for bringing this to our attention. We made this paragraph the beginning paragraph in our discussion (line 343-348). We have added additional information on implementation efforts in other realms of PGx in the discussion and highlighted how this is the first effort to implement antihypertensive PGx, including a multi-institutional approach for a small number of genes.

More specific comments:

The majority of participating sites were using similar EHRs (EpicCare Ambulatory EHR) whereas one site was using Cerner solutions. The authors write that the designed CDS was therefore adapted to EpicCare, while the site usings Cerner Solutions did not yet achieve functional implementation of CDS BPA. Further, several sites using EpicCare Ambulatory chose to manually deploy alert content rather than using Epic. What were the reasons for this?

Thank you for this comment. Since the original submission, this has changed and the Cerner site has achieved functional independence.  This has been revised around line 311 in which it is mentioned the Cerner site has achieved implementation.  As such, the paragraph in the discussion around Cerner and EHR specific limitations has been removed.

The manuscript would provide additional value for implementers if more concrete guidance could be given. Some examples include:
The authors emphasise the importance of discrete lab results reporting (p. 4). What does this specifically refer to and how was this implemented by the labs? Can you give guidance on this to other implementers who might face the same issue?

Thank you for your comment. Discrete data facilitates the alerts firing off this specific values or text as compared to full text requiring complex parsing or natural language processing.  By discrete results we mean short succinct finite observation values with only consisting of “G1/G1”, “G1/G2”, “G2/G2” for an APOL1 genotype being discretely sent through an HL7 interface in the OBX segment in addition to usual full text reports. We had discussions with the labs to ensure that the labs did not input these values in a free text field prone to user input error as deviations from the expected genotypes (“G1/G1” or “G1/G2” or “G2/G2”) would be unacceptable. We have increased the explanation in the methods and materials as to what we mean by discrete results by adding the example described. See lines 142-144 and 153-155.

Are there any open source tools or resources available that could be of use to other implementers? E.g., the patient handouts?

Thanks for this comment. We do have an open-source IGNITE toolbox through the NIH which most of our resources are available. We will be uploading this document as well as provider and patient handouts utilized in the GUARDD trial after network agreement on the addition of appropriate messaging and instructions for implementers who wish to utilize these tools. The general location for the IGNITE toolbox is below: https://dcricollab.dcri.duke.edu/sites/NIHKR/Pages/IGNITEToolbox.aspx  
The discussion would benefit from framing the implementation in relation to other PGx CDS implementation projects and frameworks.

Thanks. We discuss this above.

The font size of Figure 1 and Figure 3 is a bit too small. At the top of Figure 3, ‘Text’ should be removed?
Generally, the resolution and appearance of the figures could be improved. For Figure 3, some text is overlapping (‘Thiazide order entered…’).

Thanks. We discuss this above.

The abbreviation ‘HTN’ is used in the figures and text but not explained.

Thanks. We have defined HTN (line 255)

This manuscript is a resubmission of an earlier submission. The following is a list of the peer review reports and author responses from that submission.

Round 1

Reviewer 1 Report

The major criticisms for this paper are associated with:

  1. discussion is very poor, presenting only basis data without deeply analysing of the problem and comprehensive the authors` data with the observation form another researchers. The authors should write about strengths and weaknesses of their work.
  2. in the introduction section, the aim of the study was not clearly described. The authors have to emphasize the importance of the problem under study and the research conducted so far in this area.
  3. The authors should use the template step by step. English has to be checked by English native speaker.
  4. The passive voice should be use instead the active voice in a whole manuscript.
  5. the number of clinical groups is insufficient for discussion at this journal level.
  6. the figures are of poor quality

Reviewer 2 Report

The research article by Thomas et al "Multi-institutional implementation of clinical decision support for APOL1, NAT2, and YEATS4 genotyping in antihypertensive management" is translational and the study findings can contribute to guide the optimal selection of antihypertensive medications among an African American hypertensive population. Further, The CDS data from multiple centers strengthen the successful integration of APOL1, NAT2, and YEATS4 genomic data in antihypertensive management.

I strongly support the publication of these findings as they can be helpful in personalizing the antihypertensive treatment. I have minor comments:

  1. Please check the alignment of the text and boxes in figure 1.
  2. Line 146-check for the typo.

Reviewer 3 Report

The paper by Thomas Schneider et al. is sound and interesting.

Nevertheless some issues need to be addressed before a possible publication.

Major issues:

  • What I feel missing in the study is a pharmacoeconomic evaluation. Since the cost of multiple (i.e. APOL1, NAT2, and YEATS4) genotyping is not chicken feed, I believe this should be thoroughly discussed.
  • The authors are native speakers, nevertheless English should be improved since some mistakes are obvious throughout the text, and need to be corrected. So does the punctuation.
  • Reference section should include a very recent review that is lacking (Waltzman, J.; Lin, J. Opin. Nephrol. Hypertens. 2019, 28, 375-382, about CVD connections) imho. Needless to say, this paper should be considered in the Introduction or Discussion sections.

Minor issues:

  • General text: I can’t understand why the authors put the citations after the relative punctuation marks. This system appears odd to me.
  • I suggest to avoid the term “microalbumin/creatinine ratio” and to replace with the correct “albumin/creatinine ratio”.
  • Abstract, rows 35,36: the use of numbers in brackets is confusing since it mixes with the other numbers in brackets which introduce the different parts of the abstract itself.
  • How to use this template, rows 50-56: this should be deleted.
  • Introduction, row 74: #3 needs a couple of square brackets, I suppose.
  • Introduction, row 77: “improved mortality” sounds wrong. I suggest “reduced mortality” instead.
  • Introduction, row 87,88: the word “expression” is repeated, the sentence should be modified; “of” is lacking, may be.
  • & Meth., row 153: “a clear necessary” what this means?
  • Results, rows 184,189: “Fires” with the upper case seems wrong.
  • Results, row 236: there is an excess comma.
  • Results, row 241: “acknowledge”: is there a “d” missing? (ackowledged).
  • Results, row 268: “was change”, is there a “d” missing? (was changed).
  • Results, rows 285,286: this statement is wrong. The authors should add: “provided that non-pharmacologic actions are already implemented”, or something similar.
  • Results, row 287: there is an excess “that”.
  • Results, row 298: there is an excess comma, and perhaps a “that” is lacking.
  • Results, rows 327,328: The sentence is too long to be clear.
  • Author contributions, rows 391-395: it is confusing and should be re-written and specified.
  • References: the doi code is lacking in many entries and should be added.